# Association between metabolic surgery and cardiovascular outcome in patients with hypertension: A nationwide matched cohort study

**Erik Stenberg**[1]*, **Yang Cao**[2], **Richard Marsk**[3], **Magnus Sundbom**[4], **Tomas Jernberg**[5], **Erik Näslund**[3]

**1** Department of Surgery, Faculty of Medicine and Health, Örebro University, Örebro, Sweden, **2** Clinical Epidemiology and Biostatistics, School of Medical Sciences, Örebro University, Örebro, Sweden, **3** Division of Surgery, Department of Clinical Sciences, Danderyd Hospital, Karolinska Institutet, Stockholm, Sweden, **4** Department of Surgical Sciences, Uppsala University, Uppsala, Sweden, **5** Division of Cardiovascular Medicine, Department of Clinical Sciences, Danderyd Hospital, Karolinska Institutet, Stockholm, Sweden

* erik.stenberg@regionorebrolan.se

**Data Availability Statement:** Data cannot be shared publicly because of patient confidentiality under current Swedish legislation. Data are available from the Scandinavian Obesity Surgery

## Abstract

### Background

Hypertension, together with obesity, is a leading cause of mortality and disability. Whilst metabolic surgery offers remission of several metabolic comorbidities, the effect for patients with hypertension remains controversial. The objective of the present study was to evaluate the effect of metabolic surgery on cardiovascular events and mortality on patients with morbid obesity (body mass index [BMI] $\geq$ 35 kg/m$^2$) and hypertension.

### Methods and findings

We conducted a matched cohort study of 11,863 patients with morbid obesity and pharmacologically treated hypertension operated on with metabolic surgery and a matched non-operated–on control group of 26,199 subjects with hypertension (matched by age, sex, and area of residence) of varied matching ratios from 1:1 to 1:9, using data from the Scandinavian Obesity Surgery Register (SOReg), the Swedish National Patient Registers (NPR) for in-hospital and outpatient care, the Swedish Prescribed Drug Register, and Statistics Sweden. The main outcome was major adverse cardiovascular event (MACE), defined as first occurrence of acute coronary syndrome (ACS) event, cerebrovascular event, fatal cardiovascular event, or unattended sudden cardiac death. The mean age in the study group was 52.1 ± 7.46 years, with 65.8% being women (n = 7,810), and mean BMI was 41.9 ± 5.43 kg/m$^2$. MACEs occurred in 379 operated-on patients (3.2%) and 1,125 subjects in the control group (4.5%). After adjustment for duration of hypertension, comorbidities, and education, a reduction in risk was seen in the metabolic surgery group (adjusted hazard ratio [HR] 0.73, 95% confidence intervals [CIs] 0.64–0.84, P < 0.001). The surgery group had lower risk for ACS events (adjusted HR 0.52, 95% CI 0.41–0.66, P < 0.001) and a tendency towards lower risk for cerebrovascular events (adjusted HR 0.81, 95% CI 0.63–1.01, P = 0.060)

Registry (contact via www.ucr.uu.se/soreg/) for researchers who meet the criteria for access to confidential data.

**Funding:** ES received grants from Region Örebro County (Grant number: OLL-884791) and the Bengt Ihre Foundation. EN received grants from Stockholm County Council, SRP Diabetes, and the NovoNordisk Foundation. The funders had no role in study design, data collection and analysis, decision to publish, or preparation of the manuscript.

**Competing interests:** The authors have declared that no competing interests exist.

**Abbreviations:** ACS, acute coronary syndrome; BMI, body mass index; CI, confidence interval; COPD, Chronic Obstructive Pulmonary Disease; HR, hazard ratio; IQR, interquartile range; IRR, incidence rate ratio; MACE, major adverse cardiovascular event; NPR, National Patient Registers; SOReg, Scandinavian Obesity Surgery Register; T1DM, type 1 diabetes; T2DM, type 2 diabetes.

compared with controls. The main limitations with the study were the lack of information on BMI and history of smoking in the control group and the nonrandomised study design.

## Conclusion

Metabolic surgery on patients with morbid obesity and pharmacologically treated hypertension was associated with lower risk for MACEs and all-cause mortality compared with age- and sex-matched controls with hypertension from the general population.

---

## Author summary

### Why was this study done?

- Hypertension, particularly in combination with morbid obesity, is a leading cause of mortality and disability worldwide.

- There is compelling evidence that metabolic surgery on patients with type 2 diabetes and morbid obesity leads to reduced risk for acute cardiovascular events, but the effect on patients with hypertension and morbid obesity remains unclear.

- The main purpose of this study was to evaluate whether metabolic surgery is associated with altered risk for major adverse cardiovascular events for patients with hypertension and morbid obesity compared with a matched control group from the general population.

### What did the researchers do and find?

- The nationwide, register-based study included 11,863 patients and 26,199 controls.

- Metabolic surgery was associated with reduced risk for major adverse cardiovascular event, in particular acute coronary events.

### What do these findings mean?

- This study indicates that metabolic surgery (gastric bypass and sleeve gastrectomy) is associated with reduced risk for major adverse cardiovascular events for patients with hypertension and morbid obesity, suggesting that metabolic surgery should be considered in the treatment of patients with hypertension and morbid obesity.

## Introduction

Obesity is currently one of the major global health threats [1–3]. More than 700 million people in the world are estimated to have obesity, and in 2025, the worldwide prevalence is estimated to reach 18% amongst adult men and 21% amongst adult women [1, 4]. In several Caribbean

and Middle East countries, the prevalence is already close to 50% in women [4]. As part of a metabolic syndrome, morbid obesity is associated with high rates of hypertension, dyslipidaemia, and impaired glucose tolerance, factors causing excess risk for cardiovascular events and premature mortality [1, 2, 5]. Metabolic surgery has been shown to reduce the risk for many serious sequelae of morbid obesity [6], and in many randomised trials and observational studies, there is abundant evidence for treating type 2 diabetes (T2DM) with metabolic surgery [7, 8].

Hypertension, together with high body mass index (BMI), is a leading cause of mortality and disability [3]. A combination of the two may also be associated with resistant hypertension and increased risk for organ damage [9]. Compared to T2DM, the effect of metabolic surgery on hypertension is less well studied. Observational studies suggest that remission rates of hypertension as high as 40% can be expected at midterm follow-up after metabolic surgery [10]. The only randomised trial evaluating the effect of metabolic surgery on hypertension is a small single-centre study demonstrating remission of hypertension in 51% of the surgically treated patients [11]. The effect of metabolic surgery on cardiovascular events and mortality remains unclear in patients with hypertension and morbid obesity.

The aim of the present study was to evaluate the effect of metabolic surgery on cardiovascular events and mortality in patients with morbid obesity and hypertension.

## Methods

The Scandinavian Obesity Surgery Register (SOReg) is a nationwide register for bariatric and metabolic surgery, containing virtually all patients operated with metabolic surgery in Sweden since 2007 [12]. Data from this register were used to identify patients operated with primary gastric bypass or sleeve gastrectomy for morbid obesity. Patients younger than 18 years were excluded from the study. A 1:10 matched group of non-operated–on individuals, based on age, sex, and regional area of residence in Sweden, was created using Statistics Sweden by exact matching. Both cohorts (58,007 operated-on patients and 580,070 matched controls) were cross-linked with the Swedish National Patient Registers (NPR, based on ICD-10 diagnoses) for in-hospital and outpatient care [13], the Cause-of-Death Register [14], the Swedish Prescribed Drug Register (based on ATC codes) [15], and Statistics Sweden (https://www.scb.se). All follow-ups and definitions were relative to the date of surgery for both the intervention group and the matched controls. Subjects without hypertension and those with antihypertensive therapy possibly for other reasons were excluded from the study. Other reasons for such therapy included previous diagnosis of heart failure (ICD-10: I50) or cardiomyopathy (ICD-10: I42) treated with loop-diuretics (ATC code: C03C); heart failure treated with a beta-blocker (ATC code: C07AB02; C07AB07; C07AG02), ACE inhibitor (ATC code: C09A; C09B), or angiotensin II inhibitor (ATC code: C09C); and previous diagnosis of atrial fibrillation, flutter, or other tachycardia (ICD-10: I47 and I48) treated with a beta-blocker (ATC code: C07) or calcium antagonist (ATC code: C08D). Patients without at least 1 matched control with hypertension were also excluded from the study. This resulted in a matched cohort study with a varied matching ratio ranging from 1:1 to 1:9, including a group of 11,863 patients with hypertension operated on with metabolic surgery and a non-operated–on control group of 26,199 subjects with hypertension (Fig 1).

Although an original study plan was decided on by the authors, it was not officially documented beforehand. In response to peer review, duration of hypertension was added as a covariate, and higher education was considered as 1 group instead of being divided into 2 categories. Stratified analyses of specific subgroups (any comorbidity, T2DM, previous acute coronary syndrome [ACS], and BMI lower or higher than 40 kg/m$^2$) were also added as post hoc analyses. The STROBE reporting guideline was used to guide reporting in the paper (S1 Table).

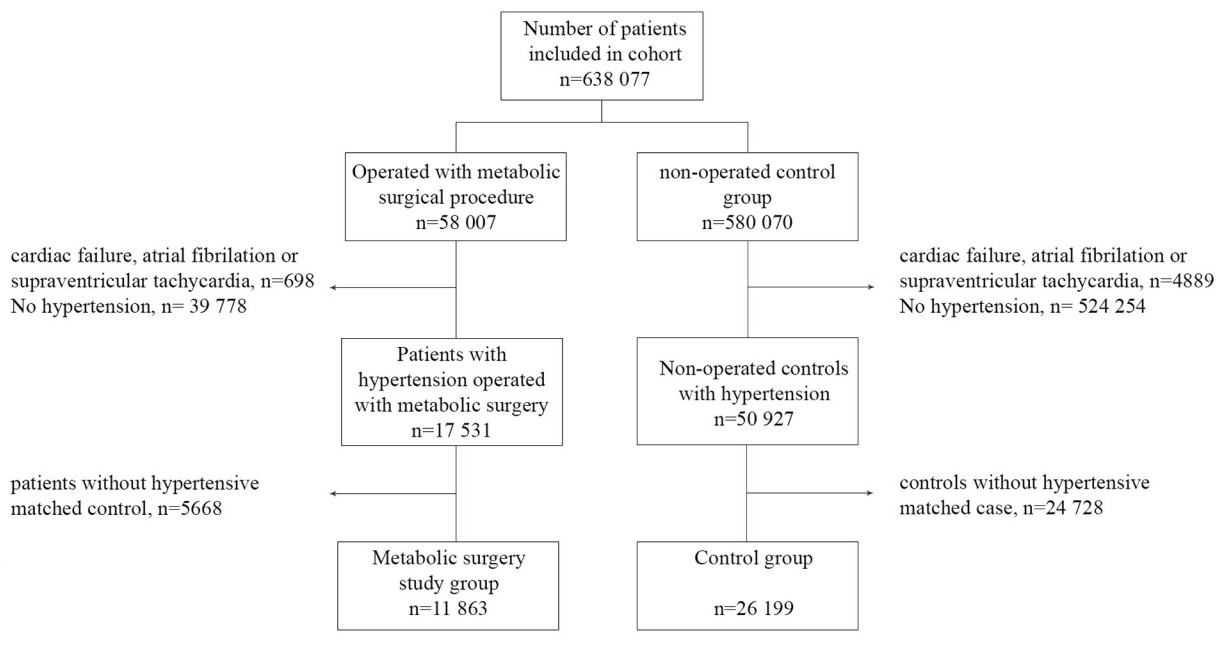

**Fig 1. Flow chart.** Flow chart describing study inclusion.

## Definition of covariates

Hypertension was defined as being prescribed antihypertensive medication (ATC code: C02, C03, C07, C08, or C09) within 18 months prior to surgery. Dyslipidaemia was defined as being prescribed medication for hyperlipidaemia (ATC code: C10) within 18 months prior to surgery. T2DM was defined as being prescribed antidiabetic medication (ATC code: A10) within 18 months prior to surgery. Patients prescribed insulin (ATC code: A10A) and diagnosed with type 1 diabetes (T1DM) in the NPR for in-hospital or outpatient care were considered to have T1DM. Chronic Obstructive Pulmonary Disease (COPD) was defined as admission for COPD or a complication of COPD with COPD as secondary diagnosis in the NPR for in-hospital care (ICD-10: J44) or prescription of an anticholinergic drug (ATC code: R03BB), long-acting beta-2 agonist (ATC codes: R03AC12–R03AC18), or a combination of these (ATC code: R03AL) indicating moderate to severe COPD [16]. Cerebrovascular disease was defined as subarachnoid haemorrhage (ICD-10: I60), intracerebral haemorrhage (ICD-10: I61), ischaemic stroke (ICD-10: I63), or acute cerebral event not specified as haemorrhage or ischaemia (ICD-10: I64) registered in the NPR for in-hospital or outpatient care. Previous ACS was defined as acute myocardial infarction (ICD-10: I21–I22) or unstable angina (ICD-10: I20.0) registered in the NPR for in-hospital care.

Sleep apnoea was defined as registration of sleep apnoea (ICD-10: G47.3) in the NPR for in-hospital or outpatient care.

Level of education was based on data from Statistics Sweden on highest completed education and classified as primary school (up to 9 years), secondary school (10–12 years of completed education), or higher education.

## Procedure

The surgical technique for laparoscopic gastric bypass is highly standardised in Sweden, with the majority being an antecolic, antegastric, Roux-en-Y gastric bypass with a small (<25 mL) gastric pouch, an alimentary limb of 100 cm, and a biliopancreatic limb of 50 cm [17]. The sleeve

gastrectomy procedure is less standardised but routinely performed using a 32–36 Fr bougie, starting resection no more than 5 cm from the pylorus, and ending 1 cm lateral to the angle of His.

## Outcome

The main outcome was a major adverse cardiovascular event (MACE), defined as first occurrence of ACS (unstable angina or myocardial infarction), cerebrovascular event (subarachnoid haemorrhage, intracerebral haemorrhage, ischaemic stroke, or acute cerebrovascular disease not specified as haemorrhage or ischaemia), fatal cardiovascular event (cause of death ICD-10: I01–78, excluding I30), or unattended sudden cardiac death (ICD-10: R96.0, R96.1, R98, and R99) registered in the NPR for in-hospital care or the Cause-of-Death Register.

Secondary outcome measures were specific cardiovascular events defined as first episode of ACS, cerebrovascular event, all-cause mortality, cardiovascular mortality, and remission of hypertension. Remission of hypertension was defined as not being prescribed an antihypertensive between 24 and 48 months after surgery for subjects with follow-up of at least 48 months.

## Statistical methods

Categorical values were presented as numbers and percentages, and continuous values as mean ± standard deviation or median with interquartile range (IQR) as appropriate. Time to first episode of MACE and time to death for all-cause mortality were estimated and visualised using the Kaplan–Meier method and presented as cumulative probability (Nelson–Aalen estimate). Cox regression for the matched cohort study was used to estimate hazard ratios (HRs) and corresponding 95% confidence intervals (CIs) for MACEs, ACS events, acute cerebrovascular events, all-cause mortality, and mortality for cardiovascular events. The chance of reaching remission of hypertension was estimated using both unadjusted and adjusted conditional Poisson regression with incidence rate ratios (IRRs) and 95% CIs as measures of association. All Cox regression and conditional Poisson regression analyses were both unadjusted as well as adjusted for comorbidity (dyslipidaemia, T1DM and T2DM, COPD, previous acute coronary event, previous cerebrovascular event, and sleep apnoea), duration of hypertension, and level of education. Proportional hazards assumption was tested using Schoenfeld residuals for all variables, and no violation was found. Potential risk factors related to remission of hypertension in the metabolic surgery group were also evaluated using a Poisson regression model, further adjusted for age, sex, surgical method, and excess BMI loss [(Initial BMI − postoperative BMI)/(Initial BMI − 25)]. Missing data were handled by listwise deletion.

IBM SPSS version 25 (IBM, Armonk, NY, USA) and Stata version 16.0 (StataCorp, College Station, TX, USA) were used for all statistical analyses.

## Ethics

The study was approved by the Regional Ethics committee in Stockholm (ref: 2013/535-31/5, 2017/857-32, and 2018/664-32) and followed the standards of the 1964 Helsinki Declaration and its later amendments. No written consent was obtained from the study participants. However, in accordance with Swedish legislation, all participants were informed of the research and quality registry and that the data would be used in clinical research, giving the patients the right to deny participation.

## Results

Compared to the control group, patients with metabolic surgery were slightly younger and more often had dyslipidaemia, diabetes, COPD, and sleep apnoea but a slightly lower incidence of cerebrovascular disease (Table 1).

**Table 1. Baseline characteristics.**

| | Operated-on Group, N = 11,863 | Non-Operated–on Control Group, N = 26,199 |
|---|---|---|
| **Age, mean ± SD** | 52.1 ± 7.46 | 54.6 ± 7.12 |
| **BMI, kg/m$^2$, mean ± SD** | 41.9 ± 5.43 | – |
| **Sex** | | |
| Male, n (%) | 4,053 (34.2%) | 9,338 (35.6%) |
| Female, n (%) | 7,810 (65.8%) | 16,861 (64.4%) |
| **Education[1]** | | |
| Primary education (≤9 years), n (%) | 2,237 (18.9%) | 4,998 (19.2%) |
| Secondary education (10–12 years), n (%) | 6,795 (57.5%) | 13,079 (50.2%) |
| Higher education, n (%) | 2,791 (23.6%) | 7,951 (30.5%) |
| **Duration of hypertension** | | |
| <1 year, n (%) | 1,599 (13.5%) | 3,090 (11.8%) |
| 1–2 years, n (%) | 1,175 (9.9%) | 2,859 (10.9%) |
| >2 years, n (%) | 9,089 (76.6%) | 20,250 (77.3%) |
| **Comorbidities** | | |
| Dyslipidaemia, n (%) | 4,437 (37.4%) | 7,802 (29.8%) |
| T2DM, n (%) | 3,328 (28.1%) | 2,690 (10.3%) |
| T1DM, n (%) | 676 (5.7%) | 911 (3.5%) |
| COPD, n (%) | 467 (3.9%) | 571 (2.2%) |
| Previous acute coronary event, n (%) | 531 (4.5%) | 1,209 (4.6%) |
| Cerebrovascular disease, n (%) | 274 (2.3%) | 877 (3.3%) |
| Sleep apnoea, n (%) | 1,789 (15.1%) | 363 (1.4%) |

[1]Missing information on highest completed education for 40 subjects in the operated-on group and 171 in the non-operated–on control group. There were no missing values for any of the remaining variables. **Abbreviations:** BMI, body mass index; COPD, Chronic Obstructive Pulmonary Disease; T1DM, type 1 diabetes; T2DM, type 2 diabetes

Amongst operated-on patients, 10,692 (90.1%) underwent a gastric bypass procedure and 1,171 (9.9%) a sleeve gastrectomy. In total, 11,428 operations were completed with laparoscopic technique (96.3%), 301 were primarily open procedures (2.5%), and 134 were converted to open surgery (1.1%). Mean follow-up time was 61.1 ± 30.4 months (1,834 ± 913 days) amongst operated-on patients and 60.7 ± 30.6 months (1,820 ± 918 days) for the non-operated–on group. Mean BMI before surgery was 41.9 ± 5.43 kg/m$^2$ in the surgery group.

## MACEs

An ACS event, cerebrovascular event, or cardiovascular death occurred in 379 operated patients (cumulative incidence at 3,000 days, 5.5%), and 1,125 subjects in the control group (cumulative incidence at 3,000 days, 7.3%) during the follow-up period. An MACE occurred for 17 patients operated on with sleeve gastrectomy (cumulative incidence at 3,000 days, 8.9%) and 362 patients operated on with gastric bypass (cumulative incidence at 3,000 days, 5.4%). Compared with the nonsurgical patients, the risk for an MACE was reduced by approximately one-fourth (unadjusted HR = 0.73, 95% CI 0.65–0.82, P < 0.001) in the metabolic surgery group. Cumulative hazard and adjusted risk for an MACE are presented in Fig 2 and Table 2, respectively. In a subgroups analysis, patients with BMI < 40 kg/m$^2$ (unadjusted HR 0.86, 95% CI 0.71–1.04, P = 0.121; adjusted HR 0.73, 95% CI 0.58–0.92, P = 0.007), as well as those with

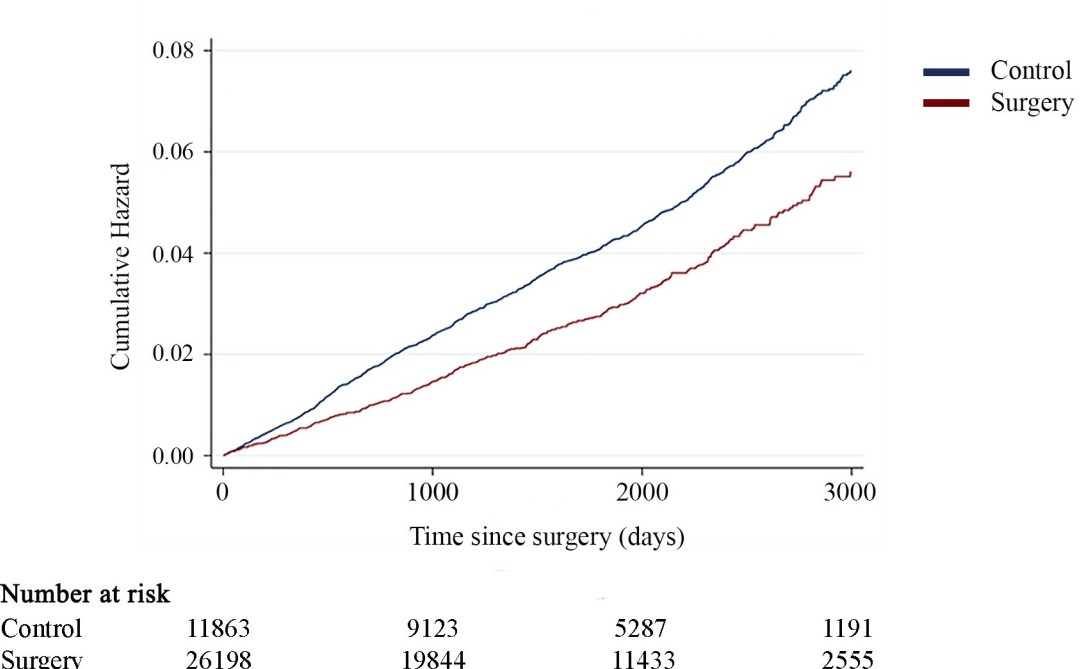

**Number at risk**

| | | | | |
|---|---|---|---|---|
| Control | 11863 | 9123 | 5287 | 1191 |
| Surgery | 26198 | 19844 | 11433 | 2555 |

**Fig 2. Cumulative hazard for MACEs.** Cumulative hazard for an MACE comparing patients with hypertension operated on with metabolic surgery compared with a matched control group with hypertension from the general population (adjusted HR = 0.73, 95% CI 0.64–0.84, P < 0.001). CI, confidence interval; HR, Hazard ratio; MACE, major adverse cardiovascular event.

**Table 2. Adjusted risk for MACEs.**

| | HR (95% CI) | Adjusted P | P for Trend* |
|---|---|---|---|
| Metabolic surgery | 0.73 (0.64–0.84) | <0.001 | |
| Dyslipidaemia | 0.98 (0.84–1.13) | 0.766 | |
| T2DM | 1.37 (1.15–1.63) | <0.001 | |
| T1DM | 2.72 (2.12–3.49) | <0.001 | |
| COPD | 1.88 (1.36–2.60) | <0.001 | |
| Previous acute coronary event | 2.29 (1.83–2.87) | <0.001 | |
| Cerebrovascular disease | 2.47 (1.93–3.16) | <0.001 | |
| Sleep apnoea | 0.92 (0.70–1.21) | 0.547 | |
| **Duration of hypertension** | | | 0.154 |
| <1 year | Reference | Reference | |
| 1–2 years | 1.19 (0.87–1.62) | 0.271 | |
| >2 years | 1.21 (0.95–1.53) | 0.122 | |
| **Education** | | | |
| Primary education (≤9 years) | Reference | Reference | |
| Secondary education (10–12 years) | 0.83 (0.72–0.97) | 0.020 | |
| Higher education | 0.66 (0.53–0.83) | <0.001 | |

Conditional Cox regression model evaluating risk for MACEs including all variables presented in the Table. **Abbreviations:** CI, confidence interval; COPD, Chronic Obstructive Pulmonary Disease; HR, hazard ratio; MACE, major adverse cardiovascular event; T1DM, type 1 diabetes; T2DM, type 2 diabetes

*No linear trend was found for using duration of hypertension as continuous variable. Likelihood ratio test was not statistically significant between the models using duration of hypertension as categorical variable and as continuous variable (P = 0.528).

BMI $\geq$ 40 kg/m$^2$ (unadjusted HR 0.78, 95% CI 0.67–0.92, P = 0.003; adjusted HR 0.71, 95% CI 0.58–0.85, P < 0.001), experienced lower risk for MACEs compared with the control group.

Metabolic surgery resulted in a significantly reduced risk for an ACS event (HR 0.61, 95% CI 0.50–0.75, P < 0.001; adjusted HR 0.53, 95% CI 0.42–0.67, P < 0.001), whilst the effect of metabolic surgery on cerebrovascular events did not reach significance (HR 0.90, 95% CI 0.75–1.09, P = 0.283; adjusted HR 0.81, 95% CI 0.66–1.01, P = 0.063).

## Mortality

In all, 472 patients (cumulative incidence at 3,000 days, 8.0%) in the operated-on group died during the follow-up period and 1,197 in the control group (cumulative incidence at 3,000 days, 8.6%); 108 patients died of a cardiovascular cause in the surgery group (cumulative incidence at 3,000 days, 2.1%) and 283 in the control group (cumulative incidence at 3,000 days, 1.9%). Metabolic surgery was associated with lower risk for all-cause mortality after adjustment for comorbidity and level of education (unadjusted HR 0.98, 95% CI 0.87–1.10, P = 0.760; adjusted HR 0.84, 95% CI 0.73–0.97, P = 0.017). No association was seen in the risk for cardiovascular mortality (unadjusted HR 1.02, 95% CI 0.81–1.29, P = 0.849; adjusted HR 0.94, 95% CI 0.71–1.25, P = 0.682).

## Remission of metabolic comorbidity

A total of 10,090 patients and 22,064 controls were available for evaluation of hypertension remission rates. Amongst patients operated on with metabolic surgery, 30.7% (n = 3,096) did not take medication for hypertension at any time 2–4 years after surgery, compared to 9.2% (n = 2,034) in the control group (unadjusted IRR for achieving remission 2.82, 95% CI 2.66–3.98, P < 0.001; IRR for remission after adjustment for age, sex, comorbidity, duration of hypertension, and level of education 3.26, 95% CI 3.04–3.50, adjusted P < 0.001). In a subgroups analysis, patients operated on with metabolic surgery with BMI < 40 kg/m$^2$ (unadjusted IRR 3.04, 95% CI 2.78–3.33, P < 0.001; adjusted IRR 3.72, 95% CI 3.32–4.17, P < 0.001), as well as those with BMI $\geq$ 40 kg/m$^2$ (unadjusted IRR 2.67, 95% CI 2.47–2.88, P < 0.001; adjusted IRR 2.97, 95% CI 2.71–3.25, P < 0.001), had higher chances of remission of hypertension compared with the control group.

Data on weight trajectories 2 years after surgery were available for 6,594 patients (67.1% of potentially available patients), showing a mean excess BMI loss of 76.3 ± 24.6%, a mean BMI loss of 12.5 ± 4.5 kg/m$^2$, and total weight loss of 29.5 ± 8.9%.

Higher age, dyslipidaemia, T1DM, previous ACS, and longer duration of hypertension were all associated with lower chance of remission of hypertension, whilst higher postoperative excess BMI loss was associated with higher chance of remission (Table 3).

Before surgery, median numbers of antihypertensive drugs were 2 (IQR 1–3) in the surgery group (mean numbers of drugs 2.3 ± 1.18) and 2 (IQR 1–2) in the control group (mean numbers of drugs 1.9 ± 1.01). After surgery, the median number of drugs was reduced to 1 (IQR 0–2) in the surgery group (mean number of drugs 1.4 ± 1.29), whilst no major difference was seen in the control group (median number of drugs 2, IQR 1–3, mean 2.0 ± 1.22) (Fig 3).

In total, 1,571 operated patients at the 4-year follow-up had been able to discontinue pharmacological treatment for dyslipidaemia (41.6%), compared to 827 (12.5%) of control subjects with dyslipidaemia at baseline (P < 0.001). Of those with T2DM at baseline, 1,806 operated patients (63.7%) were able to discontinue their pharmacological treatment, compared with 77 (3.5%) in the control group (P < 0.001) at the 4-year follow-up.

**Table 3. Adjusted IRRs for reaching remission of hypertension.**

| | IRR (95% CI) | P | P for Trend† |
|---|---|---|---|
| Age | 0.98 (0.97–0.98) | <0.001 | |
| Male sex | 0.97 (0.88–1.08) | 0.589 | |
| Dyslipidaemia | 0.80 (0.71–0.90) | <0.001 | |
| T2DM | 1.04 (0.93–1.17) | 0.495 | |
| T1DM | 0.75 (0.57–0.97) | 0.027 | |
| COPD | 0.83 (0.64–1.07) | 0.151 | |
| Previous ACS | 0.49 (0.33–0.73) | <0.001 | |
| Cerebrovascular disease | 0.84 (0.56–1.27) | 0.406 | |
| Sleep apnoea | 0.97 (0.84–1.11) | 0.688 | |
| **Duration of hypertension** | | | <0.001 |
| <1 year | Reference | Reference | |
| 1–2 years | 0.83 (0.72–0.95) | 0.008 | |
| >2 years | 0.44 (0.40–0.50) | <0.001 | |
| **Education** | | | |
| Primary education (≤9 years) | Reference | Reference | |
| Secondary education (10–12 years) | 0.95 (0.84–1.08) | 0.439 | |
| Higher education | 1.02 (0.89–1.18) | 0.756 | |
| Excess BMI loss | 2.15 (1.78–2.58) | <0.001 | |
| **Surgical method** | | | |
| Gastric bypass | Reference | Reference | |
| Sleeve gastrectomy | 0.84 (0.67–1.06) | 0.141 | |

Poisson regression model including all variables in the Table. **Abbreviations:** ACS, acute coronary syndrome; BMI, body mass index; CI, confidence interval; COPD, Chronic Obstructive Pulmonary Disease; IRR, incidence rate ratio; T1DM, type 1 diabetes; T2DM, type 2 diabetes

†A linear trend was found for using duration of hypertension as continuous variable. Likelihood ratio test was not statistically significant between the models using duration of hypertension as categorical variable and as continuous variable (P = 0.080).

## Discussion

Metabolic surgery was associated with lower risk for an MACE and overall mortality amongst patients with hypertension and obesity at baseline compared to a matched control group with hypertension from the general population. The main effect appeared to be a reduction of ACS events.

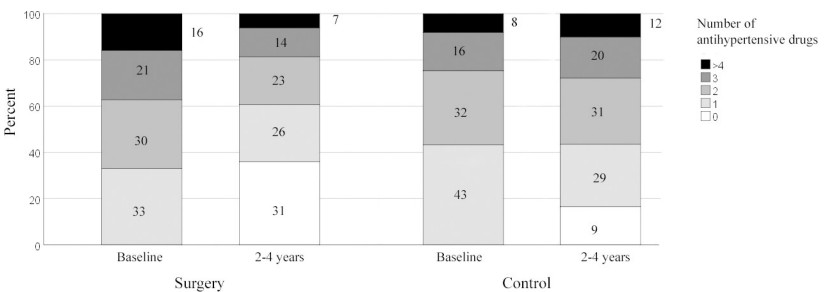

**Fig 3. Stacked histogram of numbers of antihypertensive drugs before and 2–4 years after surgery.** Stacked histogram for numbers of antihypertensive drugs before surgery and 2–4 years after surgery for hypertensive patients operated on with metabolic surgery compared with a matched control group from the general population with hypertension.

This reduction in risk is in line with that reported for the metabolic surgery group in general as well as for patients with T2DM [18–21]. A recent publication by Aminian and colleagues reported a reduction in the risk for MACEs and mortality in patients with T2DM that underwent metabolic surgery [20]. Comparison of effects of treatment in different populations and studies should be made with caution. Even so, data in the present study suggest that hypertensive patients in the surgery group experienced slightly lower reduction in the risk for an MACE as well as overall mortality compared to those with T2DM at baseline [20]. Yet, the preventive effect on ACS events appears to be higher in patients with hypertension compared with those with T2DM [20]. In contrast to the small risk reduction for cerebrovascular events reported for patients with T2DM [20], we could only demonstrate a tendency towards reduced risk for cerebrovascular events for the operated hypertensive patients. The major risk reduction for patients with hypertension thus appears to be more cardiovascular than cerebrovascular, which is in agreement with previous reports for patients with T2DM and morbid obesity [22]. Patients who suffer an ischaemic cerebrovascular event may, however, have fewer poststroke complications and better disability status, as demonstrated by Han and colleagues [23]. In addition to well-established risk factors and surgery or not, higher education was also associated with reduced risk for an MACE. This supports the previous findings that the risk for coronary heart disease is increased in patients with a low level of education, which was reported in a large mendelian randomisation study [24].

Because hypertension (in particularly in combination with obesity) is a major risk factor for an acute coronary event, remission of hypertension would seem a likely explanation for this preventive effect. Although the remission rate of hypertension was well in line with the rates reported for similar groups of patients [10, 25, 26], the 31% remission rate at 4 years in the present study was lower than that reported in the only randomised clinical trial to date on the subject, the GATEWAY trial [11]. The GATEWAY trial, however, was conducted on a healthier group of patients with lower BMI compared with the present study. The discontinuation and true remission of hypertension has recently been questioned [27], and indeed, relapse of hypertension amongst patients with early remission has been reported to be high [28]. The main benefit of metabolic surgery for patients with hypertension may thus not be remission of hypertension itself, but rather a combination of protective cardiometabolic effects. The effects of bariatric surgery on glucose metabolism and T2DM are well documented [7, 8, 20, 25], and the high remission rates of T2DM in the present study supports this. Furthermore, metabolic surgery has been reported to reduce the thickness of the media wall and pulse wave velocity, in particularly in patients with dyslipidaemia and hypertension [29]. Although dyslipidaemia was associated with lower chance of achieving remission of hypertension, the improvement in important comorbidities such as dyslipidaemia and T2DM signals a general improvement in overall cardiometabolic control after metabolic surgery. Metabolic surgery has also been reported to reduce general systemic inflammation as well as the development of atherosclerosis [30, 31]. With weight reduction being an important measure to prevent and treat hypertension, metabolic surgery may also transform the situation of the patient with poor response or resistance to pharmacological treatment for hypertension into a more benign situation, reducing the risk for development of organ damage [32, 33]. Many of these protective effects are likely to contribute to the lower risk for cardiovascular events as well as overall mortality seen in the present study.

In contrast to the higher hypertension remission rate seen after gastric bypass compared with sleeve gastrectomy in the SLEEVEPASS study [34] (51% for gastric bypass and 29% for sleeve gastrectomy), no difference was seen in the chance of achieving hypertension remission between gastric bypass and sleeve gastrectomy in the present study. Study groups in randomised clinical trials, however, are often highly selected, providing results that are more

impressive than those seen in routine clinical practice [35, 36]. Moreover, the present study was not designed to compare the 2 surgical methods, which is why this result should be viewed with caution. Increasing age, dyslipidaemia, T1DM, and previous acute coronary events were all associated with lower chances of achieving remission of hypertension. As with remission of T2DM, patients with shorter duration of disease and those losing more weight had a higher chance of remission [37].

Metabolic surgery may not be the global solution to the obesity epidemic. However, with high blood pressure and obesity being two of the leading causes of mortality and morbidity worldwide [3], metabolic surgery should be considered an important part of treatment if available.

### Strengths and limitations

This study was based on a large, nationwide cohort of patients with hypertensive controls matched by age, sex, and place of residence. Follow-up rates for the major outcomes were very high thanks to excellent registration in the national high-quality registers used. The main limitations, however, were the lack of data on BMI and history of smoking in the control group and the fact that this was not a randomised study. Despite the limitations of BMI as a measure of obesity and predictor of outcome after bariatric surgery [38], it remains an important part in the evaluation of candidates for bariatric surgery and is associated with increased mortality [39]. Although we have compensated for other known risk factors in the statistical evaluations, it is likely that the overall healthier patients in the control group may result in underestimation of treatment effects in the surgery group.

The study relied on pharmaceutical use in the definition of hypertension and comorbid disease, without accurate data on blood pressure levels at follow-up. Although all drugs evaluated in the study are prescribed (and thus included in the registers), we have no data on prescribed drugs not being taken by the patient. Noncompliance with recommended treatment is not uncommon even when treatment for disease such as hypertension is so important [40]. The remission rates (9.2% for hypertension and 12.5% for dyslipidaemia) in the control group may well represent noncompliance with medical treatment. Furthermore, duration of hypertension was introduced in the study as an amendment to the original study plan. We did not have data from a long enough period to evaluate separate, longer durations of hypertension. Although shorter duration was associated with higher chance of remission, the effects of longer duration could thus not be evaluated.

Finally, whilst the Swedish Cause-of-Death Register is generally regarded to be a high-quality register with virtually complete data, the historically low autopsy rates in Sweden may make definite cause of death more difficult to ascertain [14]. This could explain why no difference in cardiovascular mortality was seen between the groups despite the lower incidence of ACS events and all-cause mortality in the surgical group.

## Conclusion

Metabolic surgery in patients with morbid obesity and hypertension decreases the risk for MACEs and all-cause mortality compared with age- and sex-matched hypertensive controls from the general population.

## Supporting information

**S1 Table. STROBE checklist.** STROBE, Strengthening the Reporting of Observational Studies in Epidemiology.
(DOC)

**S1 Fig. Cumulative risk for MACEs, stratified by T2DM.** For study participants with T2DM, cumulative incidence at 3,000 days was 6.8% for the surgery group and 10.3% for the control group. For study participants without T2DM, cumulative incidence at 3,000 days was 5.0% for the surgery group and 7.0% for the control group. MACE, major adverse cardiovascular event; T2DM, type 2 diabetes.
(TIF)

**S2 Fig. Cumulative risk for MACEs, stratified by presence of comorbidity other than hypertension.** For study participants with comorbidity, cumulative incidence at 3,000 days was 6.7% for the surgery group and 11.2% for the control group. For study participants without comorbidity, cumulative incidence at 3,000 days was 3.7% for the surgery group and 5.0% for the control group. MACE, major adverse cardiovascular event.
(TIF)

**S3 Fig. Cumulative risk for MACEs, stratified by preoperative ACS.** For study participants with previous ACS, cumulative incidence at 3,000 days was 18.1% for the surgery group and 18.8% for the control group. For study participants without previous ACS, cumulative incidence at 3,000 days was 4.8% for the surgery group and 6.8% for the control group. ACS, acute coronary syndrome; MACE, major adverse cardiovascular event.
(TIF)

## Author Contributions

**Conceptualization:** Erik Stenberg, Magnus Sundbom, Tomas Jernberg, Erik Näslund.

**Data curation:** Erik Stenberg, Yang Cao.

**Formal analysis:** Erik Stenberg, Yang Cao.

**Funding acquisition:** Erik Stenberg, Erik Näslund.

**Investigation:** Erik Stenberg, Yang Cao.

**Methodology:** Erik Stenberg, Yang Cao, Tomas Jernberg, Erik Näslund.

**Project administration:** Erik Stenberg, Magnus Sundbom, Erik Näslund.

**Resources:** Erik Stenberg, Richard Marsk, Magnus Sundbom, Erik Näslund.

**Software:** Yang Cao.

**Supervision:** Erik Näslund.

**Validation:** Erik Stenberg, Yang Cao.

**Visualization:** Erik Stenberg.

**Writing – original draft:** Erik Stenberg, Yang Cao.

**Writing – review & editing:** Yang Cao, Richard Marsk, Magnus Sundbom, Tomas Jernberg, Erik Näslund.

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
