## [Editor Report · Decision Letter 0]

4 May 2020

Dear Dr Stenberg, 

Thank you for submitting your manuscript entitled "Association between metabolic surgery and cardiovascular outcome in patients with hypertension: a nationwide matched cohort study" for consideration by PLOS Medicine.

Your manuscript has now been evaluated by the PLOS Medicine editorial staff [as well as by an academic editor with relevant expertise] and I am writing to let you know that we would like to send your submission out for external peer review.

Kind regards,

Adya Misra, PhD,

Senior Editor

PLOS Medicine

---

## [Decision Letter · Decision Letter 1]

4 Jun 2020

Dear Dr. Stenberg,

Thank you very much for submitting your manuscript "Association between metabolic surgery and cardiovascular outcome in patients with hypertension: a nationwide matched cohort study" (PMEDICINE-D-20-01660R1) for consideration at PLOS Medicine. 

[LINK]

In light of these reviews, I am afraid that we will not be able to accept the manuscript for publication in the journal in its current form, but we would like to consider a revised version that addresses the reviewers' and editors' comments. Obviously we cannot make any decision about publication until we have seen the revised manuscript and your response, and we plan to seek re-review by one or more of the reviewers. 

We expect to receive your revised manuscript by Jun 25 2020 11:59PM. Please email us (plosmedicine@plos.org) if you have any questions or concerns.

We look forward to receiving your revised manuscript. 

Sincerely,

Emma Veitch, PhD

PLOS Medicine

On behalf of:

Adya Misra, PhD

Senior Editor 

PLOS Medicine

plosmedicine.org

*In the last sentence of the Abstract Methods and Findings section, please briefly note some of the key limitation(s) of the study's methodology.

*It would be good to clarify in the paper if the analyses reported here were planned out prospectively in a pre-existing protocol or analysis plan? Please state this (either way) early in the Methods section.

*In the Methods section it would be good to add a note that the STROBE reporting guideline was used to guide reporting in the paper and there to call out the supporting information file that corresponds to the STROBE checklist. 

Comments from the reviewers:

Reviewer #1: Association between metabolic surgery and cardiovascular outcome in patients with hypertension: a nationwide matched cohort study shows not only new, but very interesting data about a controversial issue in Metabolic Surgery, the impact on hypertension.

I have some comments and I hope they can improve your manuscript.

MAJOR

1. Lack of data on BMI is a critical issue about your manuscript. Revising other papers based on Sweden's registers, they showed the control group's BMI. Why do you not have it?

2. Data about smoking would be very important. Do you have it?

MINOR

1. In the abstract, the adjustment was made for more than one comorbidity. You should correct to comorbidities.

2. In methods: "The Scandinavian Obesity Surgery Register (SOReg) is a nationwide register for metabolic surgery" I suggest including Bariatric and Metabolic surgery.

3. When you cite ICD-10 and ATC-codes in the Definition of covariates, you should make clear from which coding system they are.

4. In the Outcome, do you think that it is necessary to include the ICD-10 codes? I suggest deleting them to make the text clearer.

5. Table 1 should include the BMI. 

6. The follow-up time should be expressed in months. You can maintain days between parentheses. 

7. Table 2 is a little confusing for me. Patients with T2DM submitted to Metabolic Surgery, for example, have more risk of MACE than control groups?

8. Although it is not the objective of your manuscript, can you show the other causes of mortality?

9. After the first appearance of type-1 and type-2 Diabetes Melitus you should use T1DM and T2DM that are frequently used in many manuscripts.

10. References: The 3-years results from GATEWAY were presented at AHA and the abstract is published, and you can include the results: 

Schiavon CA, Bhatt DL, Santucci EV, Oliveira JD, Santos RN, Damiani LP, Machado RH, Noujaim PM, Halpern H, Monteiro FJ, Sousa MG, Amodeo C; Bortolotto L, Ikeoka DT, Cavalcanti AB, Berwanger O, Drager LF. Effects of Bariatric Surgery in Patients With Hypertension: 3-Year Outcomes From the Randomized GATEWAY Trial. Circulation. 2019;140:e972. DOI: 10.1161/CIR.0000000000000742.

Sincerely,

Carlos Aurelio Schiavon

Reviewer #2: Thank you very much for the opportunity to review this manuscript. 

Overall, this is an interesting manuscript investigating a relevant topic for many patients, The paper is in line with many prior publications of the Scandinavian Obesity Surgery Register linked with national healthcare databases. 

The manuscript is well-written, nicely prepared and clearly presented. It further supports the value of metabolic surgery in the treatment of a variety of comorbidties of metabolic diseases. However, there are some major issues with the underlying data (registry based and linking of many national databases using ICD-codes) which are acknowledged by the authors and can unfortunately not be addressed. Despite these limitations, I support the acceptance of the paper after revisions. The value of this paper is that it investigates for the first time patients specifically with hypertension independent of underlying comorbidities except obesity in a large cohort. 

1. Also, if you have a relevant number of patients with a BMI <35kg/m2, could you provide the outcome stratified for these low-BMI patients? This questions would be of interesting to know and provide some answers if metabolic surgery is also of value in patients with BMI <35kg/m2. If you do not have these data, than you may be able to make an analysis comparing patients with BMI > and <40kg/m2. I do think that the notion that excess weight loss is associated with more hypertension remission should be related to with the preoperative BMI because EWL depends on the preoperative weight. 

2. Could you provide a subgroup analysis for the following subgroups to better define which patients benefit as outlined below including figures like Fig. 2? It would be interesting to know whether one of the subgroup benefits in particular from metbalic surgery or if all benefit equally. 

a. Patients with or without diabetes? 

b. Patients with prior cardiac incident (prior MACE) 

c. Patients without any known comorbidities 

Reviewer #3: I confine my remarks to statistical aspects of this paper. The general approach is fine but I have some issues to resolve before I can recommend publication.

p. 5 Why was a 1 to 10 matching chosen? How was the matching done? Much more detail is needed. (There are tons of ways to do matching)

p. 6 There is a typo People with 10 years of education don't fall in any category. 

 Why this division for education? In particular, why < 3 vs. > 3 years of higher ed? Are college degrees usually 3 years in Sweden? Why not degree vs. not?

p 10 Don't categorize years of hypertension. Categorizing continuous variables is nearly always a mistake. In *Regression Modeling Strategies* Frank Harrell lists 11 problems with this and sums up "nothing could be more disastsrous" Instead, leave years as years and use a spline to investigate nonlinearity

 The referennce category for education needs to be specified.

In the limitations, you should note the problems with BMI as a measure of obesity., See e.g. this blog post I wrote: https://medium.com/peter-flom-the-blog/why-bmi-is-a-bad-measure-of-obesity-and-what-is-better-f8a62fc9ca49?source=friends_link&sk=4b2ea559ab12853beb577764f83d151a

Figure 3 Don't use stacked bar charts (see the work of William S. Cleveland). I think a mosaic plot might be good here, but it depends on what you are trying to show.

Peter Flom

[LINK]

---

## [Decision Letter · Decision Letter 2]

9 Jul 2020

Dear Dr. Stenberg,

Thank you very much for re-submitting your manuscript "Association between metabolic surgery and cardiovascular outcome in patients with hypertension: a nationwide matched cohort study" (PMEDICINE-D-20-01660R2) for review by PLOS Medicine.

I have discussed the paper with my colleagues and the academic editor and it was also seen again by xxx reviewers. I am pleased to say that provided the remaining editorial and production issues are dealt with we are planning to accept the paper for publication in the journal.

[LINK]

We look forward to receiving the revised manuscript by Jul 16 2020 11:59PM. 

Sincerely,

Adya Misra, PhD

Senior Editor 

PLOS Medicine

plosmedicine.org

Requests from Editors:

Abstract

Background-please add a bit more context to illustrate the importance of this study

Please provide participant demographics 

Please change “morbidly obese” to “patients with morbid obesity” in line with the principles of people first language. The same goes for “hypertensive” which should be revised to “with hypertension”

Author summary

Please change hypertensive to patients with hypertension

Line 90 probably doesn’t need a question mark

Comments from Reviewers:

Reviewer #1: Association between metabolic surgery and cardiovascular outcome in patients with

hypertension: a nationwide matched cohort study.

I would like to thank the authors for the responses and the improvement of the manuscript.

I still have some questions to you:

1. I understood that you do not have power to compare the two surgical techniques, but can you show the data about ACS events for each technique?

2. I still have a problem with Table 2. You answered me that it is an analysis of the entire cohort, but the legend says matched cohort, what for me represents the control group. Please revise it. 

3. The definition of remission of a comorbidity is a controlled disease without an active treatment. You did not show the data about blood pressure and, because of that, I am a little concerned about the remission rate based only in patients who were not taking medications. If you have the data about blood pressure, you should show it. If not, I think it is better to address this issue in the limitations.

4. I still think that showing the others causes of mortality will enrich your paper, but I understood your point and I will let this decision to the Editor.

5. Unfortunately, I did not find the supplement Figures in the manuscript.

Carlos Aurelio Schiavon

Reviewer #2: Thank you for the revised manuscript. My comments and the other comments have been well addressed in my opinion. 

However, there are 2 minor points: 

1. I was not able to find the supplemental figures in the files

2. I think there is a labeling mistake in Fig. S4, second paragraph: I think it should be "without previous acute coronary syndrome" instead of type-2-diabetes. 

Reviewer #3: The authors have addressed my concerns and I now recommend publication

Peter Flom

[LINK]

---

## [Editor Report · Decision Letter 3]

30 Jul 2020

Dear Dr. Stenberg, 

On behalf of my colleagues and the academic editor, Dr. Carlos Schiavon, I am delighted to inform you that your manuscript entitled "Association between metabolic surgery and cardiovascular outcome in patients with hypertension: a nationwide matched cohort study" (PMEDICINE-D-20-01660R3) has been accepted for publication in PLOS Medicine. 

PRODUCTION PROCESS

PRESS

PROFILE INFORMATION

Thank you again for submitting the manuscript to PLOS Medicine. We look forward to publishing it. 

Best wishes, 

Adya Misra, PhD

Senior Editor 

PLOS Medicine

plosmedicine.org